# Dementia Care Nursing for Apathetic Older Patients: A Qualitative Study

**DOI:** 10.3390/geriatrics9050106

**Published:** 2024-08-23

**Authors:** Mana Doi, Asumi Tanaka, Nanae Nemoto, Tenna Watanabe, Yuka Kanoya

**Affiliations:** 1Department of Gerontological Nursing & Home Care Nursing, Chiba Faculty of Nursing, Tokyo Healthcare University, Chiba 273-8710, Japan; 2Department of Gerontological Nursing, Nursing Course, School of Medicine, Yokohama City University, Yokohama 236-0004, Japan; asumi.tanaka@ai-org.jp (A.T.); nanae09010614@yahoo.co.jp (N.N.); t0315cute@icloud.com (T.W.); ykano@yokohama-cu.ac.jp (Y.K.)

**Keywords:** apathy, nursing, dementia, aged

## Abstract

The number of patients hospitalized with dementia is increasing, but one symptom, apathy, tends to be overlooked and unaddressed. Thus, this study determines how nurses certified in dementia nursing engage with older patients with dementia who exhibit apathy during hospitalization. A qualitative study using semi-structured interviews with 10 dementia care nurses in Japan was conducted. Through conventional content analysis, 10 categories were generated. They included (1) initiating patient engagement when their physiological or daily-life problems become more pronounced, (2) assessing and identifying the causes of decreased motivation from multiple perspectives, (3) assessing patients from multiple perspectives to determine the best way to start supporting them, (4) providing reassurance through basic dementia care, (5) incorporating pleasant stimuli into the hospital environment, (6) providing care based on patients’ circumstances and abilities by collaborating with multiple professionals. Nurses initiate involvement with patients when their daily life problems become more pronounced. They conduct comprehensive assessments from multiple perspectives and collaborate with other professionals to ensure patient care and safety. They also extend their support to patients’ families and maintain long-term involvement. Apathetic older patients benefit from basic nursing care practices and a patient-centered approach, which do not require specialization or additional costs and resources.

## 1. Introduction

The percentage of the total population in Japan that comprises older adults has been steadily increasing, reaching 29.1% in 2023 [1]. This has been accompanied by a rising number of older adults with dementia, projected to reach approximately 7 million by 2025 [2]. Approximately 20% of patients admitted to hospitals’ general wards exhibit symptoms of dementia or cognitive decline [3]. Furthermore, the hospitalization rate of people aged 65 years and above is higher (73.2%) than that of other age groups [4]. Consequently, medical staff are inevitably required to provide more efficient care to dementia patients who are admitted to general wards.

Hospitalization represents an unfamiliar environment for older individuals with dementia. Irregular routines, medical procedures, and the distress associated with the disease often lead to heightened anxiety and stress, exacerbating the behavioral and psychological symptoms of dementia (BPSD) [5]. While research suggests that appropriate responses, such as person-centered care and non-drug interventions, can alleviate BPSD [6], nurses frequently encounter difficulties when communicating with patients who struggle with communication, verbal abuse, violence, and other challenging behaviors associated with BPSD [7]. 

Apathy, recognized as a component of BPSD [8], is defined in the International Classification of Diseases 11th Revision as “a reduction or lack of feeling, emotion, interest, or concern; a state of indifference” [9]. One study reported a 54% prevalence of apathy in persons with dementia [10], with sex, severity of dementia, personality, midlife motivational abilities, and baseline levels of apathy as influencing factors [11]. In addition to drug therapy with donepezil, non-drug treatments such as cognitive stimulation, reminiscence groups, and music and art therapies have been suggested as effective for apathy [11,12]. Non-pharmacological treatments are given priority for BPSD, including apathy, which is an issue for hospitals that do not provide such non-pharmacological treatments. 

Prior studies examining care for BPSD have reported a tendency to provide less proactive care to patients exhibiting apathy compared with those displaying behaviors such as wandering and violence, which are considered problematic for treatment [13]. Consequently, patients with apathy may not receive adequate attention, raising concerns about potential declines in their activities of daily living and the worsening of BPSD due to prolonged bed rest.

Recently, nurses with specialized knowledge and skills in specific specialties are increasingly expected to assume important roles in addressing complex patient issues, including apathy. In Japan, the introduction of the Certification in Dementia Nursing (certified nurses in dementia nursing: DCNs) as a recognized nursing specialization in 2004 has placed these nurses at the forefront of managing dementia-related symptoms and proposing tailored communication methods and decision-support strategies corresponding to dementia stages [14]. Investigation into the professional practices of DCNs [15] reveals that they engage deeply with patients with dementia by dedicating time to observing their words and actions, adjusting the care environment, and supporting patients’ families to ensure patients are cared for when they leave the hospital. DCNs also create opportunities for staff to reflect on their own nursing care. Therefore, we can reasonably presume that DCNs maintain a vigilant focus on the words and actions of older patients with dementia exhibiting apathy to safeguard their mental and physical well-being and that of their families and to guide healthcare peers. However, no prior study has described DCN involvement in the care of older patients with dementia exhibiting apathy in detail. Older patients with dementia exhibiting apathy are admitted to wards of various departments and are provided care by DCNs who specialize in dementia and work across multiple wards. It is necessary to clarify the details of their involvement at first. However, considering the increase in the number of patients, the support of DCNs alone will not be sufficient in the future, and general nurses will also need to consider how to respond to them. Therefore, this study aimed to clarify how DCNs care for older patients with dementia exhibiting apathy during hospitalization and discuss nursing care that can be provided to this population even in hospitals that do not have DCNs.

## 2. Materials and Methods

### 2.1. Design

Our study adopted an exploratory-descriptive qualitative design using content analysis.

### 2.2. Sampling and Recruitment

Participants comprised DCNs certified by the Japanese Nurses Association who had experience working with older patients with dementia exhibiting apathy in a hospital setting. We excluded DCNs who had no experience working with older patients/persons with dementia exhibiting apathy.

In Japan, certified nurses, including DCNs, are specialized nurses who need at least five years of practical experience in general (with at least three of five years in the certified nursing field). DCNs are responsible for managing various dementia symptoms, including BPSD, suggesting communication methods, providing decision-making support, and offering psychological and social support to families. Participants were recruited using the snowball sampling method, as we wanted to take advantage of the strong network among DCNs. This method allowed us to introduce early informants to the next set of participants [16]. We concluded data collection when data saturation was achieved, signifying no additional data provision regarding nursing practice [17].

### 2.3. Data Collection 

We conducted semi-structured interviews using the Zoom application, securing access with a password and identification to include only researchers and participants. The interviews were recorded using Zoom’s recording function, and the interview data were stored on a password-protected USB, which was kept in a locked cabinet. All interviews were conducted by the researchers (MD, AT, NN, or TW). To ensure the quality of the interviews, the researchers, comprising those with experience in the field of nursing care of older patients or in qualitative research, reached a consensus by reviewing the interview recordings, including the speed of speech and how to conduct an interview with several anticipated reactions. Each interview, conducted between 17 October and 2 December 2022, lasted 32–61 min. We asked various questions during the interviews (Table 1), covering various attributes, including age, sex, affiliation, nursing experience, DCN experience, days worked as a DCN, and frequency of involvement with patients exhibiting apathy. The interview questions were set based on prior studies [7,13,15], and their validity was discussed with researchers and practitioners specializing in gerontological nursing. Participants not currently working in wards were asked to recall their experiences in that setting.

### 2.4. Data Analysis 

We employed conventional content analysis as described by Hsieh and Shannon [18], following the inductive approach outlined in Elo and Kyngäs [19]. After the interviews were transcribed, we extracted data related to nursing care as described by the participants. The contents of the extracted data were reviewed and abstracted. The abstracted data were coded, ensuring that the participants’ meaning was retained. These codes were then abstracted into sub-categories and categories and named based on the type of nursing care, emphasizing the similarities and differences in meanings. The research team, comprising certified nurse specialists in gerontology, researchers with geriatric nursing experience, and researchers well-versed in qualitative research, reviewed the entire analysis process to ensure data reliability.

### 2.5. Ethical Considerations

The participants were fully informed about the study’s purpose prior to participation, and written consent was obtained. This study was approved by Yokohama City University Ethics Committee (approval number: F220800014).

## 3. Results

Ten participants were included in the study (Table 2). The analysis resulted in 297 codes, grouped further into 10 categories and 35 sub-categories, as summarized in Table 3. The 10 categories of nursing care were divided as follows: three pertained to initiating patient engagement, four were regarding care methods, and three were related to continued engagement and support for patients and their families. The italicized sentences refer to these categories, along with their corresponding category numbers in parentheses. Representative quotes from participants are included below.

### 3.1. Initiating Patient Engagement

Three categories relating to the timing of the initiation of care engagement were generated. DCNs initiated direct involvement with patients when problems in the patients’ daily lives became noticeable. They assessed the patients from different perspectives and explored ways to support them accordingly.

DCNs *initiate patient engagement when their physiological or daily life problems become more pronounced (1)*. That is, they obtain/accept consultation from the staff when a patient’s physiological and daily life problems become more noticeable (sub-category 11).

“When apathy is mentioned, I first think of loss of appetite. For instance, a patient may have been talking normally and energetically upon admission but became unresponsive after surgery. Despite thorough examinations, nothing appears wrong, prompting the staff to inquire ‘What is happening with them?’” (DCN 7) 

DCNs *assess and identify the causes of decreased motivation from multiple perspectives (2)*. That is, they assess the causes of low motivation by considering the patients’ mental and physical conditions (sub-category 5) and collaborate with multiple professionals to assess patients if they show signs of apathy, hypoactive delirium, or depression (sub-category 8).

“I strive to discern whether the decline in activities of daily living stems from physical conditions or mental factors such as inability to focus or take actions to fulfill one’s needs”. (DCN 3)

“Do they really not want to call us, do they not want to be involved with us, or are they unable to call us?… I always try to pay attention to those who cannot speak up”. (DCN 10)

“First, they are treated for hypoactive delirium, after which the physical symptoms calm down and the delirium improves, but if the lethargy or lack of motivation persists, then it’s assumed that apathy is still present. Differentiation is also very difficult”. (DCN 1)

DCNs *assess patients from multiple perspectives to determine the best way to start supporting them (3)*. They explore the patient’s life and preferences prior to their hospitalization from diverse perspectives to determine the required support (sub-category 9).

“Before hospitalization, I gather information about the patient’s preferences and habits regarding eating. For those who tire easily or require assistance with eating, I use this knowledge to facilitate mealtimes effectively”. (DCN 10) 

### 3.2. Care Methods for Patients 

Four categories of care methods emerged from the information provided by DCNs: *providing reassurance through basic dementia care (4), incorporating pleasant stimuli into the hospital environment (5), providing care based on patients’ circumstances and abilities by collaborating with multiple professionals (6),* and *administering basic nursing care, extending beyond addressing apathy (7)*. They worked with multiple professionals to provide patients with basic care with an emphasis on safety and comfort without giving up.

DCNs *provide reassurance through basic dementia care (4)*. They continue to actively talk to the patient and explain the situation so that the patient feels secure during the care (sub-category 20).

“I try to communicate with patients, especially after completing care, by expressing gratitude for their cooperation or explaining forthcoming medical procedures such as starting an intravenous drip. I encourage our staff to do the same”. (DCN 1) 

DCNs *incorporate pleasant stimuli into the hospital environment (5)*. They use devices, photos, and visits by family members to stimulate the patients (sub-category 24).

“I encourage the inclusion of items such as letters, message cards, photos, or other personal mementos to strengthen the patient’s connection with their family. Such stimuli can improve their daily life. I often request family members to bring these mementos because I think they might have positive effects”. (DCN 1)

DCNs *provide care based on patients’ circumstances and abilities by collaborating with multiple professionals (6)*. They provide care using the patient’s strengths and abilities (sub-category 14).

“I try to ensure timely consultations with physical and occupational therapists to identify welfare equipment that enhances the patient’s independence. I believe that a single piece of equipment can make a significant difference”. (DCN 5) 

DCNs *administer basic nursing care, extending beyond addressing apathy (7).* They adjust the basic rhythm of patients’ lives using available opportunities and adjust the environment for leaving the bed, sitting during the daytime and communicating more (sub-category 10).

“I would like to ask staff to tap patients’ shoulders at a certain time in the morning, even if their eyes are closed. In addition, I do not want to force patients to sit and eat if they are tired, but I would still like staff to try asking patients to do so because it is important for them to sit and eat”. (DCN 8)

“We also consider how to get them out of bed; of course, we do general things, such as getting them in a wheelchair for a short time and getting them some morning sunlight”. (DCN 10)

### 3.3. Continued Engagement and Support for Patients and Their Families

Three categories were generated regarding post-nursing care and family support. After approaching the patient, the nurses assessed and corrected their reactions and also provided physical and mental support for the family.

DCNs *evaluate patients’ responses and modify the nursing approach accordingly (8)*. They employ certain strategies to engage, involve, and encourage patients to respond, even if the response is minuscule (sub-category 1).

“I closely monitor patients’ eating behavior and gauge their reactions. Even in the depths of apathy, when asked about their preferences, they display subtle signs, such as eye movement, nodding, or gesturing. I consider their past favorite foods and suggest starting with those”. (DCN 3) 

DCNs *maintain long-term involvement with patients without giving up (9).* They establish a true relationship by visiting the patient’s room repeatedly and talking to them even if they do not respond (sub-category 27). They are involved with patients while keeping in mind that it is difficult to show marked improvement despite implementing the approach (sub-category 13).

“I value smiles and greetings! I try to visit patients at different times throughout the day. Even a brief appearance can make a difference, particularly if they have been in bed since lunchtime. At times like that, I go (to the patient’s bed) and say, ‘I’m here again’; often, I come back and say, ‘Please listen to me’, or ‘I have something to say’. Even if I just show my face a little, I think it makes a difference; therefore, I want to create a relationship where they feel like it is okay to talk to them for a while”. (DCN 8)

“Well, no matter how much support we give, we need to understand that it won’t necessarily be successful… we need to understand that this is the normal state of the person”. (DCN 3)

DCNs *also support the physical and mental well-being of patients’ families by collaborating with multiple professionals (10)*. They cooperate with multiple professionals to provide mental and physical support to the family (sub-category 35).

“Caring for the patient extends beyond patient care; family is equally vital. The family’s emotional and physical well-being impacts the patient, so we have to care about them. We ask about their well-being and sleep quality and, if necessary, connect them with other professionals for support”. (DCN 5) 

“Patients lose their spontaneity. When that happens, it’s very hard on the family caring for them, so they hold these gatherings to let off some steam and talk about it. I think it would be better if there were more outlets for the family members to release their stress… I can talk with them at the outpatient clinic”. (DCN 6)

## 4. Discussion

As indicated by category 1, DCNs *initiate patient engagement when their physiological or daily life problems become more pronounced (1)*. Disruptive behaviors such as aggression among BPSD require immediate action and intervention because they cause great distress to both patients and caregivers [20]. Based on our results, when deciding the best time to begin engaging with patients with dementia, nurses should consider not only disruptive behaviors but also instances of reduced responsiveness or decreased activity without an apparent physical cause, which could indicate apathy. 

Once DCNs begin their involvement with older patients with dementia exhibiting apathy, they proceed to *assess and identify the causes of decreased motivation from multiple perspectives (2)*. This shows that DCNs do not rely solely on physical characteristics to assess their patients. This is important, as loss of motivation may result from disturbances in cognition, emotion, or level of consciousness [21,22], which can be influenced by pharmacotherapy and poor physical or mental health. Additionally, we clarified that DCNs *assess patients from multiple perspectives to determine the best way to start supporting them (3)*. Given the emotional withdrawal experienced by patients with apathy, assessing the effectiveness of medical staff support is difficult. Thus, to gather evidence for supporting the patient, information collected from family members and the patient’s history before hospitalization plays a crucial role in DCNs’ understanding of the patient. As indicated in sub-category 9, DCNs investigate the patient’s life and preferences before hospitalization from different perspectives to determine the support needed.

Four categories of care methods for patients emerged from our analysis. First, DCNs *provide reassurance through basic dementia care (4)*. Rather than providing special care for apathy, they utilize fundamental approaches for dementia care, such as reality orientation and person-centered care [23], ensuring patient safety [20]. Moreover, nurses bear the responsibility of providing comfort through non-pharmacological therapies for older individuals regardless of the care setting [20]. Thus, *incorporating pleasant stimuli into the hospital environment (5)*, such as photos of patients’ families, was also determined to be a necessary care component for older patients with dementia exhibiting apathy. DCNs also *provide care based on patients’ circumstances and abilities by collaborating with multiple professionals (6).* Collaborative care is an important concept in dementia care, with physical and occupational therapists possessing detailed knowledge of patients’ abilities, functions, and mobility [23]. It has been suggested that when DCNs engage with older patients with dementia experiencing apathy, they share comprehensive information about patients’ abilities and circumstances with medical staff since it is difficult to collect information directly from the patients. Category 7 reveals that DCNs *administer basic nursing care, extending beyond addressing apathy (7)*. According to the Principles for Older Persons (United Nations) [24], regardless of apathy or dementia, older people should be supported to enable them to live with dignity and enjoy fundamental freedoms. This support includes fundamental nursing care that encourages and supports activities such as sitting to eat. Our results, emerging from category 7, suggest that DCNs consistently provide basic care, which carries the risk of being overlooked, particularly in the case of apathy, where patients may voice fewer complaints and needs. Thus, we suggest providing basic nursing care regardless of patients’ complaints or needs. 

Following their interventions, DCNs *evaluate patients’ responses and modify the nursing approach accordingly (8)*. They observe patients’ reactions, continuing with approaches that yield positive responses and making necessary corrections in case of no response. This demonstrates their commitment to refining their approach in line with patient feedback. Importantly, the effectiveness of these approaches may not be immediately evident. However, as shown in sub-category 27, “They establish a true relationship by visiting their (patients’) room repeatedly and talking to them even if they do not respond”, DCNs continued to *maintain long-term involvement with patients without giving up (9)*. Individuals with apathy may display reduced emotional responses to positive or negative events [22], rendering response generation from patients difficult. Therefore, medical staff, including nurses, must consider this aspect when engaging with older patients with dementia experiencing apathy and maintain their commitment to providing them with quality care. For patients’ family members, we clarified that DCNs *support the physical and mental well-being of patients’ families by collaborating with multiple professionals (10),* including physicians and other multidisciplinary professionals. In addition to our findings for DCN 5, “The family’s mental and physical support will impact the patient”, Merrilees et al. [25] highlighted the emotional impact of apathy on patients’ families. Taken together, these findings suggest that supporting family members is crucial to facilitate the continuation of family involvement as long as possible.

Among the categories generated, categories 1 and 9 are unique. Category 1 shows that general nurses consult with certified nurses or that certified nurses get involved when the patient’s physiology or daily life presents a serious problem. If nurses have concerns regarding eating, sleeping, or excretion among older patients with dementia and observe a worsening of mental symptoms, such as apathy, they initiate early intervention to address apathy. Category 9 emerged because we interviewed DCNs. Staff are likely to abandon further approaches once engagement with patients/persons with dementia fails repeatedly; as apathy is a low-activity condition, staff are often not bothered by it, as indicated in prior research [13]. Therefore, such symptoms are likely to go unnoticed and are not commonly addressed. However, as category 9 shows, DCNs will use their knowledge and skills to somehow improve the patient’s situation despite the patient being non-responsive or a lack of observable good results. This approach is partially based on their pride as professionals. In this regard, we found that DCNs also shared with general staff the intervention ideas that they had identified through their long-term interactions with patients, created care plans for patients, and persistently worked with them.

In contrast, some practices elucidated in this study require no specialized skills or abilities; therefore, they can be utilized by any nurses, including those with no certification in dementia nursing. When general nurses interact with people with dementia, they should remain vigilant for signs of apathy, initiate patient engagement, and actively involve themselves in their care. Additionally, collaboration with multidisciplinary teams is necessary to provide basic care that prioritizes safety and comfort. It is also necessary to support the mental and physical well-being of patients’ families. After providing some interventions, careful observation and evaluation of patient responses are crucial, with a focus on continuous improvement of the approach and not surrendering to hopelessness, if possible. 

However, this study has some limitations that should be acknowledged. First, the participants included only DCNs in Japan. In addition, our sample might be restricted to a rather small network of acquaintances because of the sampling method. Second, the patients recalled by DCNs in the interviews were patients who were thought to be apathetic and who were actually patients who needed treatment for apathy and, therefore, were not necessarily patients who had been diagnosed with apathy. Third, owing to the nature of qualitative research, generalizing the results may be challenging. To address this, we recommend recruiting a broader network of nurses, including various kinds of specialized nurses from other countries with aging populations, and adopting quantitative approaches to confirm the generation of new data, which may facilitate the generalization of the results.

## 5. Conclusions

Our findings indicate that DCNs initiate involvement with patients when their daily life problems become more pronounced, conduct comprehensive assessments from multiple perspectives, and collaborate with other professionals to ensure patient care and safety. They extend their support to the patients’ families and maintain long-term involvement with their patients. Clearly, apathetic older patients are best served with basic nursing care practices and a patient-centered approach, which do not require specialization or additional costs and resources.

## Figures and Tables

**Table 1 geriatrics-09-00106-t001:** Interview questions.

Questions
Have there been any impressive cases of older patients with dementia experiencing apathy?
Can you share your perspectives on better understanding older patients with dementia experiencing apathy? [15]
What aspects are you aware of when engaging with older patients with dementia experiencing apathy? [15]
How do you approach older people with dementia experiencing apathy to maintain and improve their activities of daily living and cognitive function? [13]
What considerations do you keep in mind when engaging with older patients with dementia experiencing apathy? [15]
What are the difficulties and problems in engaging with older patients with dementia experiencing apathy? [7]
What are your expectations from nurses in general wards? [15]

**Table 2 geriatrics-09-00106-t002:** Characteristics of the participants (n = 10).

Demographic Variables		n (%)	
Age	30–39	2	(20.0)
	40–49	3	(30.0)
	50–59	5	(50.0)
Sex	Male	4	(40.0)
	Female	6	(60.0)
Affiliation	Hospital general ward	6	(60.0)
	Ward for mental health	1	(10.0)
	Regional healthcare network office	1	(10.0)
	Clinic	2	(20.0)
Years of experience as nurses, mean ± SD (n = 9)		21.7 ± 6.7	
Years of experience as DCN, mean ± SD		6.3 ± 2.6	
Days of working as a DCN	Yes	7	(70.0)
Frequency for engaging patients experiencing apathy	More than once a week	4	(40.0)
	More than once a month	4	(40.0)
	More than once a year	2	(20.0)

DCN: Certified nurse in dementia nursing.

**Table 3 geriatrics-09-00106-t003:** Nursing practices for older patients with dementia who exhibit apathy: Categories and sub-categories.

Theme	Category (Category No.)	Sub-Category (Sub-Category No.)
Initiating patient engagement	Initiate patient engagement when their physiological or daily life problems become more pronounced (1)	Obtain/accept consultation from the staff when a patient’s physiological and daily life problems become more noticeable (11)
		Increase involvement for patients having a poor response, long hospital stay, or sharp decline in food intake (12)
	Assess and identify the causes of decreased motivation from multiple perspectives (2)	Assess the causes of low motivation by considering the patients’ mental and physical conditions (5)
		Collaborate with multiple professionals to assess patients if they show signs of apathy, hypoactive delirium, or depression (8)
	Assess patients from multiple perspectives to determine the best way to start supporting them (3)	Explore the patient’s life and preferences prior to his/her hospitalization from diverse perspectives to determine the required support (9)
		Assess the patient’s problems from multiple perspectives: physical, mental, and social (6)
Care methods for patients	Provide reassurance through basic dementia care (4)	Continue to actively talk to the patient and explain the situation so that the patient feels secure during the care (20)
		Communicate with patients using the methods of reality orientation and humanitude (21)
		Provide reality orientation by all staff members constantly (23)
		Incorporate the patient’s interests, lifestyle, and familiarity into the hospital room environment (25)
		Minimize environmental changes after discharge by informing caregivers about the patient’s responses and reactions during hospitalization (34)
	Incorporate pleasant stimuli into the hospital environment (5)	Use devices, photos, and visits by family members to stimulate the patients (24)
		Provide an environment where patients can consider aspects besides their treatment (26)
		Use media such as photographs and other objects that the person likes or treasures as a way to communicate with other staff members (19)
		Touch patients’ hands, shoulders, and back to bring their awareness to the outside world (22)
	Provide care based on patients’ circumstances and abilities by collaborating with multiple professionals (6)	Provide care using the patient’s strengths and abilities (14)
		Collaborate with rehabilitation staff to implement rehabilitation and cognicize that the patient can perform (30)
		Work on improving their food intake by considering their oral conditions, chewing ability, and food preferences by collaborating with dietitians and speech-language-hearing therapists (32)
		Intentionally involve multiple professionals and staff members with the patients (31)
		Use information from the rehabilitation staff for the patient’s assessment and care (33)
		Approach patients based on the degree of their disease and cognitive functioning (7)
	Administer basic nursing care, extending beyond addressing apathy (7)	Adjust the basic rhythm of patients’ lives using available opportunities and adjusting the environment for leaving the bed and sitting during the daytime as well as communicating more (10)
		Talk to, observe, and respond to patients, which are basic ways of interacting with patients (29)
Continued engagement and support for patients and their families	Evaluate patients’ responses and modify the nursing approach accordingly (8)	Employ certain strategies to engage, involve, and encourage patients to respond, even if the response is minuscule (1)
		Provide verbal and non-verbal reinforcements when apathetic patients respond to increase the frequency of their responses (2)
		Identify pleasant stimuli by evaluating the patient’s responses and incorporate them into the relationship (3)
		Do not force patients by engaging with them to prevent unpleasant reactions (4)
		Identify and create ways of involving the patients that motivate and energize them by assessing their reactions to the approach (15)
	Maintain long-term involvement with patients without giving up (9)	Be involved with patients while keeping in mind that it is difficult to show marked improvement despite implementing the approach (13)
		Continue to be involved with patients without giving up even if they do not respond (16)
		Patiently observe the changes in patients over time (18)
		Establish a true relationship by visiting their room repeatedly and talking to them even if they do not respond (27)
		Share time with patients to understand their thoughts and feelings even if no words are used (17)
	Support the physical and mental well-being of patients’ families by collaborating with multiple professionals (10)	Cooperate with multiple professionals to provide mental and physical support to the family (35)
		Share the patient’s responses with their family to motivate them to become involved in the care (28)

DCN: Certified nurse in dementia nursing.

## Data Availability

The datasets generated and analyzed during the current study are not publicly available because the ethical approval body does not allow it; however, they are available from the corresponding author on reasonable request.

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
