# Peer review of "Dementia Care Nursing for Apathetic Older Patients: A Qualitative Study"

_geriatrics, 2024, doi:10.3390/geriatrics9050106_

Round 1

Reviewer 1 Report (Previous Reviewer 1)

Comments and Suggestions for Authors

The authors adequately addressed the reviewers' comments. I have no further recommendations

Author Response

Thank you for your constructive feedback on our paper.

Reviewer 2 Report (Previous Reviewer 2)

Comments and Suggestions for Authors

1.      Please change the term “dementia patients” to “patients/persons with dementia” throughout the whole manuscript.

2.      The background of the study describes the roles of DCNs and points out that there is no previous evidence to support this role. Why does the purpose mention “... discuss nursing care that can be provided to this population even in hospitals that do not have DCNs"? Additionally, the recruitment criteria stated, “Participants comprised certified DCNs who had experience working with older patients with dementia.”? Please clarify the purpose or address it more precisely.

3.      Under section 2.2, the sentence “In Japan, certified nurses, including DCNs, are required to have at least five years of practical training after obtaining a nursing license (with at least three years in the certified nursing field)” is unclear. Do you mean that a certified nurse (CN) is an advanced practice nurse who needs at least five years of practical experience in general, followed by an additional three years of experience in a specialty area to become a DCN in field (like dementia)?

Author Response

RESPONSE TO REVIEWER 2 

Thank you for your comments. We have revised our manuscript accordingly. Our revisions in the manuscript are highlighted in yellow. 

  1. Please change the term “dementia patients” to “patients/persons with dementia” throughout the whole manuscript.

Response: We changed them.

  1. The background of the study describes the roles of DCNs and points out that  there is no previous evidence to support this role. Why does the purpose mention “... discuss nursing care that can be provided to this population even in hospitals that do not have DCNs"? Additionally, the recruitment criteria stated, “Participants comprised certified DCNs who had experience working with older patients with dementia.”? Please clarify the purpose or address it more precisely.

Response: We are grateful for your comment. We added the explanation regarding our aim (Page 2, Lines67–72). 

  1. Under section 2.2, the sentence “In Japan, certified nurses, including DCNs, are required to have at least five years of practical training after obtaining a nursing license (with at least three years in the certified nursing field)” is unclear. Do you mean that a certified nurse (CN) is an advanced practice nurse who needs at least five years of practical experience in general, followed by an additional three years of experience in a specialty area to become a DCN in field (like dementia)?

Response: Thank you for your comment. We revised the manuscript as follows (Page 2, Lines 85–87): “In Japan, certified nurses, including DCNs, are specialized nurses who need at least five years of practical experience in general (with at least three of five years in the certified nursing field). 

Thank you again for your constructive feedback on our paper. We trust that the revised manuscript is suitable for publication. 

This manuscript is a resubmission of an earlier submission. The following is a list of the peer review reports and author responses from that submission.

Round 1

Reviewer 1 Report

Comments and Suggestions for Authors

The present study explores how nurses certified in dementia care engage with older patients with dementia who exhibit apathy during hospitalization. It is an interesting paper, however there are several major issues that should be addressed.

Title

The title is appropriate and clearly indicates the focus of the study. However, consider making it more concise, e.g., “Dementia Care Nursing for Apathetic Older Patients: A Qualitative Study.”

Abstract

The abstract provides a good summary, but it lacks specific results. Including key findings in the abstract would give readers a quick insight into the study’s conclusions.

Methodology

   - The sampling method (snowball sampling) is appropriate but not described in sufficient detail. Explain why this method was chosen and how it was implemented.

   - The interview questions provided are relevant, but the methodology section should include more detail on the development and validation of these questions.

   - The data collection process is described briefly. Details about the interview process, how data was recorded, and steps taken to ensure confidentiality should be expanded.

   - Data analysis is mentioned, but the process needs more elaboration. Explain how codes were generated and validated and provide more details on how themes were identified.

   - Ethical approval and informed consent are mentioned but should include the name of the ethics committee that approved the study (even if it’s blinded here, it should be detailed in the final version).

Results

   - The results section is well-organized, with clear categories and sub-categories. However, it lacks depth in describing how the categories were derived from the data.

   - Include more direct quotes from participants to support the categories and sub-categories.

   - Consider adding a visual representation, such as a thematic map or table, to summarize the findings clearly.

Discussion

   - The discussion section adequately links the findings to existing literature but needs to better highlight the study's unique contributions.

   - Some of the interpretations seem to repeat what is already known from previous studies. Focus more on the novel insights gained from this study.

- A recent review (https://pubmed.ncbi.nlm.nih.gov/38015288/) addressed that one of the main concerns related to apathy is that it is often underdiagnosed and undertreated/mistreated. Even its prevalence across different studies displays high heterogeneity, possibly depending on the assessment methods. How do DCN assess apathy? Do they employ a particular screening tool? How do they make differential diagnosis with other conditions such as depression? 

   - The limitations are mentioned but should be expanded. Discuss potential biases introduced by the sampling method and the subjective nature of qualitative analysis.

   - Future research directions should be more specific. Instead of a general call for more studies, suggest specific research questions or methodological improvements.

Conclusion

   - The conclusion effectively summarizes the study but is somewhat repetitive. Condense the key points and emphasize the practical implications of the findings.

Tables and Figures

   - The tables provided are useful but could be improved. Table 3 is particularly detailed; consider breaking it into smaller, more focused tables for clarity, or changing the layout to make it more readable.

   - Adding figures to visually represent key findings or processes (e.g., a flowchart of the methodology) would enhance the readability and impact of the paper.

References

   - Ensure all references are up-to-date. Some references are older and should be replaced with more recent studies where possible.

   - Follow a consistent citation style throughout the manuscript.

The manuscript is promising but requires extensive revisions to improve clarity, depth, and readability. Focus on expanding the methodology, providing richer data in the results, and strengthening the discussion and conclusion sections. Additionally, improve the presentation of tables and consider adding figures for better visualization. By addressing these points, the manuscript will be more impactful, providing valuable insights into the specialized nursing practices for apathetic older patients with dementia.

Comments on the Quality of English Language

Moderate editing of English language required

Author Response

Thank you for your comments. We agree with you, and we revised our manuscript accordingly. Our revisions in the manuscript are highlighted in red font. Please check the response in the attachment.

Reviewer 2 Report

Comments and Suggestions for Authors

Comments on the Quality of English Language

average

Author Response

(The authors gave the same response as above.)

Reviewer 3 Report

Comments and Suggestions for Authors

My greatest concern is that the patient population that are described as apathetic is in adequately described. As a result the population reported to be apathetic could have hypoactive delirium and should be actively managed 

Author Response

(The authors gave the same response as above.)
